# Alternative Ecological Products for Aphid Control on Plum

**DOI:** 10.3390/plants12183316

**Published:** 2023-09-20

**Authors:** Claudiu Moldovan, Ioan Zagrai, Georgeta Maria Guzu, Zsolt Jakab-Ilyefalvi, Luminita Antonela Zagrai, Stefania Mirela Mang, Aurel Maxim

**Affiliations:** 1Department of Engineering and Environmental Protection, Faculty of Agriculture, University of Agricultural Sciences and Veterinary Medicine, No. 3-5, Calea Manaștur Street, 400372 Cluj-Napoca, Romania; claudiu.moldovan@usamvcluj.ro; 2Fruit Research & Development Station Bistrița, 3 Drumul Dumitrei Nou, 420127 Bistrița, Romania; ioan.zagrai@scdp-bistrita.ro (I.Z.); georgeta-maria.bivolariu@usamvcluj.ro (G.M.G.); zsolt.jakab@scdp-bistrita.ro (Z.J.-I.); luminita.zagrai@scdp-bistrita.ro (L.A.Z.); 3Department of Horticulture and Landscape, Faculty of Horticulture and Business in Rural Development, University of Agricultural Sciences and Veterinary Medicine, No. 3-5, Calea Manaștur Street, 400372 Cluj-Napoca, Romania; 4School of Agricultural, Forestry, Food and Environmental Sciences (SAFE), University of Basilicata, Viale dell’Ateneo Lucano 10, 85100 Potenza, Italy; stefania.mang@unibas.it

**Keywords:** aphids, plum, ecological system, insecticides effect, pest, ecological phytosanitary product

## Abstract

Ecological farming is increasing worldwide, as more and more consumers prefer chemical-free fruits. As a result, ecological farming is becoming increasingly appealing to European farmers, including those in Romania. However, implementing an effective ecological phytosanitary program remains a challenge for farmers due to limited options and a lack of information about their effectiveness. Romania is a major producer of plums and ranks second in the world after China. Aphids are common pests of plum, and some species are vectors of the damaging *Plum pox virus*, and therefore require special attention regarding their control. Eight ecological products were tested both in the field and laboratory for a duration of three vegetative periods to determine their efficiency in aphid control. The effects of ecological products were compared with five chemical insecticides known to be effective against aphids. Observations were made 24 and 48 h after spraying. Two of the eight ecological products tested were proven to be efficient in aphid control, Ovipron Top and Prev-Am, with a mortality rate over 90%. The results indicate that these two ecological products are comparable in effectiveness to chemical insecticides and could be suitable candidates for both ecological and conventional treatment programs.

## 1. Introduction

*Prunus domestica* L., commonly known as the plum tree, is a highly popular and valued fruit-bearing species worldwide. Its fruits are packed with vitamins and antioxidants, making it recommended to consume 2–4 portions daily for a healthy diet [1,2]. According to data from FAOSTAT, China leads in global plum fruit production with 6,465,219 tons [3]. Romania also plays a significant role in plum cultivation, ranking second in the world with a production of 757,880 tons. Plums are enjoyed fresh, processed (dried plums, jams, preserves) and distilled to produce alcoholic beverages (plum brandy) [3,4]. Cultivating plums in an eco-friendly manner is a relatively new concept globally and is currently being studied to make it a viable alternative to conventional farming methods [5]. However, plum cultivation is prone to specific diseases and pests that can severely affect fruit quality and yield.

Aphids are one of the most common pests affecting plum crops [6,7] and they mainly belong to the genus *Aphis*. They feed on the sap from leaves, causing the dwarfing of trees, premature leaf falling, and small fruits or premature fruit drop. Furthermore, the infested plants exhibit abnormal growth and young trees may even completely dry out and thus die [7,8,9].

Some species of aphids, such as *Aphis spiraecola* Patch, *Myzus persicae* Sulzer, *Brachycaudus persicae* Passerini and *Rhopalosiphum padi* Linnaeus, are vectors in the natural transmission of *Plum pox virus* (PPV) in a non-persistent manner [10,11,12,13], often causing serious stone fruit economic losses [14]. This virus is transmitted when aphids feed on infected plant sap and then move to a healthy plant, spreading the virus through feeding. This makes the monitoring and treatment of these pests a crucial step in plum orchard management. Reducing PPV spread is also important in order to establish the flight curve of aphid populations in each area so that treatments can be applied at the proper periods and with maximum effectiveness [15]. Overall, the control of aphid populations is very important in order to reduce the direct damages, and to limit the spread of PPV which produces significant additional losses. Previous field studies have shown that the application of mineral oils, which are accepted in ecological farming, is useful in reducing the incidence of PPV infections [16]. The allelopathic effect of phenolic acids can also be a good tool for protecting plants against aphid infestation; a high concentration of phenolic acid (gallic acid and caffeic acid) inflicts a reduction in aphid infestation [17]. Another alternative method to decrease aphid infestation can be the application of organic blends of volatiles which can attract the pest, allowing its capture [18].

Ecological farming systems use specific treatment targeting to replace the use of synthetic chemical products [5,19]. Europe’s focus on environmental strategies aimed at reducing pollution that harms the environment, wildlife and humans has led to an increased interest in ecological orchards [5,20]. These changes are being seen in both developed countries with advanced agriculture and developing countries that are trying to meet current market demands, with consumers being a key factor in this change.

Romania has recorded an increase in ecological orchards after implementing measure 4.1a “Investments in fruit tree orchards” under the National Rural Development Program as per the Regulations (EU) 1305/2013, 1307/2013 and the principles of the CAP 2014–2020. Ecological orchards established with non-repayable funds from the European Union under this measure received higher funding, making it a more attractive option for farmers. The issue that arises is the success rate of these orchards and how much of the forecasted production will be achieved in order to provide ecological fruits to the consumers. Plum is still the dominant fruit species in Romania [3,4] and it is desirable if even part of its production is ecological. However, controlling the main diseases and pests in orchards remains a big challenge due to limited information about the actual effectiveness of ecological phytosanitary products. Despite the fact that Romania has a significant role in the global plum production market, a major part of pest and disease control is still mainly based on chemical treatments.

The present study aimed (1) to evaluate the effectiveness, under laboratory conditions, of eight insecticidal products that are approved for use in ecological agriculture; (2) to assess the effectiveness of these products for controlling aphids when tested in the field; (3) to compare the effectiveness of the products tested in a controlled environment with those tested in the field; and (4) to evaluate the effects of ecological product treatments on the aphid mortality rate over a three-year period.

## 2. Results

### 2.1. Laboratory Testing of Ecological and Chemical Products against Aphids

In the first experiment conducted under laboratory conditions, the results obtained over the three years of study (2019–2021) revealed that after 24/48 h of treatment, two out of eight ecological products showed a high aphicidal efficiency. In particular, the highest mortality rate of insecticidal products was recorded for the Ovipron Top (conc. 0.3%) variant, which caused a 95–98% mortality rate of aphids after 48 h, followed by the Prev-Am (conc. 0.8%) variant with a mortality rate of aphids up to 90% after the same period of time. The other products tested showed weaker results regarding their efficacy in aphid control under the given conditions (Figure 1).

In the laboratory test, all the chemical synthesis products tested showed a high level of efficacy, causing a mortality rate of over 90% within 48 h after treatment. On average, their effectiveness increased by about 10% between the two observation time intervals (24/48 h). This kind of reaction was expected because these products have a systemic effect, meaning that as the pests consume the active substance through feeding, their effectiveness will progressively increase (Figure 1).

### 2.2. Field Testing of Ecological Products against Aphids

The results obtained in the field testing of some ecological and chemical products on microvariants were similar to those obtained under laboratory conditions 48 h post-treatment. The highest efficacy regarding the aphid mortality rate, after the application of ecological products with an insecticidal effect, was recorded for Ovipron Top (96%) followed by Prev-Am (90%). All the other tested products showed a reduced efficacy, similar to the results obtained under laboratory conditions 24 and 48 h after spraying (Figure 2).

In line with the laboratory results, the synthetic chemical products demonstrated a very high effectiveness also under field conditions. Their efficiency was confirmed by an aphid mortality rate of over 94% observed within 48 h post-treatment (Figure 2).

### 2.3. Data Analysis Results

Statistical analysis outcomes showed that significant differences 48 h post-treatment were registered between the tested ecological products in relation to the aphid mortality rates (Table 1).

Of the eight ecological products tested after the statistical analysis, Ovipron Top and Prev-Am were the most efficient for aphid control compared to all other products used in the study. Therefore, the previously presented results regarding percentage efficacy (Figure 1 and Figure 2) were also statistically assured. The differences between the two products (Ovipron Top and Prev-Am) and the other ecological products tested (Oleorgan, Algasil, Canelys, Konflic, Deffort and BactoSpeine DF) were significant (Table 1) and the most efficient product validated in this study was Ovipron Top, while the least efficient was BactoSpeine DF.

Following ANOVA (analysis of variance) and Duncan’s Multiple Range Test, it can be observed that there are no significant differences between the five chemical products tested (Actara, Movento, Mospilan, Karate Zeon and Calypso) and it can be assumed that they all demonstrated the same insecticidal effect against aphids, proven by the high mortality rate of aphids registered (Table 2).

The efficiencies of the tested chemical products, followed by statistical analysis, are presented in Table 3. The other chemical products, which have registered a good efficiency, can still be found on the market and are successfully applied in conventional phytosanitary treatment programs for aphid management.

Moreover, we verified whether or not there were statistically assured differences between the two periods of time (Figure 3) after the application of treatments (24/48 h) on ecological variants over the entire experimental period (2019–2021), and the results revealed that there were no significant differences between them (Table 3).

The value of *p* calculated was more than alpha = 0.05; thus, the difference between the two observations was not significant.

Following the statistical processing of data regarding the efficacy of ecological products compared with chemically synthesized ones, from the eight ecological products tested, two had shown a similar efficacy to the conventional ones (Ovipron Top and Prev-Am) 48 h post-treatment (Table 4).

The results demonstrated that following Duncan’s test (*p* < 0.001), the efficacy of the OvipronTop and Prev-Am products compared to the chemical ones was not significant. In particular, the results from the whole experimental period showed that only these two ecological products demonstrated a high aphid control efficiency, which was also statistically assured. Therefore, these two products can be ideal candidates for use in ecological plum crops, having a high efficacy in the control of aphids, comparable with consecrated chemical products. The statistically assured results demonstrated the fact that the efficiency of these two products can be observed after a relatively short time following application, precisely just 24 h post-treatment. This interval of time was sufficient to observe the beneficial effects of the products, taking into account that these are contact-type products.

## 3. Discussion

The present study represents one of the few investigations regarding the control of aphids on plum species by using ecological products with an insecticidal effect. Until now, to our knowledge, there have been no similar scientific reports published using ecological and chemical trials to control aphids for longer experimental time periods. Thus, the results from this study provide important data for both the scientific community and farmers, and also fill the actually existing gap in this domain. Experiments from the present study, performed over three vegetative periods (2019–2021), allowed for the long-term testing of different ecological products with insecticidal effects. Some of them are presented in the scientific literature for their effects in controlling other diseases and pests. For example, the ecological products Ovipron Top, Prev-Am and Deffort were applied in the control of cherry blackfly (*Myzus cerasi* Fabricius) with favorable results seen for the first two products [21]. In addition, Konflic, Deffort, and Prev-Am have been proven to be efficient in the control of *Tuta absoluta* (Meyrick) [22,23], but also in the control of powdery mildew in the case of Prev-Am [24].

Another ecological product used in our present study, Canelys, was efficient in the control of fungal diseases from the *Praghmidium* genus [25]. But, it could also have an acaricidal effect, according to the manufacturers. A high efficiency in the control of aphids in almond was reported following the application of Oleorgan and Rotorgana [26]. Bactospeine DF use was proven to have an efficient effect in controlling several families of pests from the *Lepidoptera* order [27,28].

With the increasing surfaces in the ecological system, essential oils together with other accepted components in ecological agriculture are more and more present and used [29]. Refined mineral oils are an important component of the technological chain, from different pharmaceutical domains [30]. Recent studies have shown the benefits of these oils and their applicability in the control of several diseases and pests in different species of plants [21,23,24]. The preparation formula and the obtained concentrations from the extracts of different parts of the plants turned out to be good regarding phytosanitary protection in the ecological system [31]. Thus, more and more phytosanitary products accepted in the ecological agriculture have extracts of mineral oils as a basis. Orange mineral oils are known to be efficient in controlling many diseases and especially pests. At the moment of application of a product like this, the pest’s organism, due to the concentration and the content of the oil, is affected irremediably, finally leading to their death [32]. In the frame of the current experience, it is worth noting that some of the chemical products used have, in the meantime, been withdrawn from the market due to their toxicity and impact on the environment. Of the eight ecological products tested for their efficacy in plum aphid control, only two have been proven efficient, one having a base of paraffin mineral oil (Ovipron Top) and the other, orange mineral oil (Prev-Am). The high efficiency of these products, as shown in this study, could be expected, based on previous scientific data which supported the beneficial effects of the obtained products from mineral oils for the control of different diseases and specific pests in plants. Thus, the tested products Ovipron Top and Prev-Am are promising candidates which should be integrated into the ecological phytosanitary treatment programs for plum crops.

## 4. Materials and Methods

### 4.1. The Pedoclimatic Conditions in Which the Experiments Took Place

The research took place in Bistrița, Romania, located in the northeastern Transylvanian Plateau, surrounded by the “Hills of Bistrita”, at 47°10′ north latitude and 358 m altitude. The climate is temperate-continental with an average annual temperature (last 25 years) of 10 °C, and 720 mm of annual precipitation, respectively. The summers are hot and humid, and the winters are dry and cold. The soil is eutricambosol, with medium NPK and organic matter content. The micrometeorological parameters were recorded using an Adcon Telemetry automated weather station in the orchard.

### 4.2. Sampling Material and Ecological/Chemical Products’ Origin and Use

The study was carried out at the Fruit Research and Development Station Bistrița (FRDS Bistrița) over three consecutive vegetative periods (2019–2021), with three repetitions per year and focused on microvariants of the plum species (*Prunus domestica* L.). The experiment was divided into two parts: one carried out in a laboratory setting, and the other in the field. A total of eight ecological insecticides were tested and compared with five conventional chemical products commonly used by farmers against aphid infestations in plum crop. The treatments were applied to infested shoots using the manufacturer-recommended concentrations according to the following formula:$$\mathrm{Amount}\;\mathrm{of}\;\mathrm{insecticide}\;\left(\mathrm{mL}\;\mathrm{or}\;g\right)\;\mathrm{per}\;\mathrm{liter}\;\mathrm{of}\;\mathrm{water}=\frac{\mathrm{quantity}\;\mathrm{of}\;\mathrm{water}\;\mathrm{we}\;\mathrm{need}}{\mathrm{liquid}/\mathrm{ha}\;\left(1000\;L\right)}\times \mathrm{Concentration}\;\mathrm{required}\;\mathrm{mL},g/\mathrm{ha}$$

The aphid mortality rate was reported as a percentage after the calculation of the average of three repetitions for each variant and was recorded 24 and 48 h after treatments. The ecological insecticides tested were Konflic (Atlántica Agricola S.A., Spain), Prev-Am (Oro Agri Europe S.A., Portugal), Oleorgan (Atlántica Agricola S.A., Spain), Algasil (AltincoAgro, Spain), Canelys (Atlántica Agricola S.A., Spain), Ovipron Top (United Phosphorus Ltd., India), Deffort (AltincoAgro, Spain) and BactoSpeine DF (Nufarm, Australia). A detailed list of the ecological products applied in both laboratory and field conditions for aphid control is presented in Table 5.

The chemical insecticides used for comparison with ecological products were the following: Calypso (Bayer, Germany), Mospilan (Summit Agro, Japan), Actara (Syngenta, Switzerland), Movento (Bayer, Germany) and Karate Zeon (Syngenta, Switzerland). Details about the chemical insecticides used in this study are presented in Table 6.

### 4.3. Laboratory Testing of the Ecological and Chemical Products 

During the laboratory tests performed over a three-year vegetative period (2019–2021), thirteen phytosanitary products were evaluated in three repetitions/year, eight of them being ecological (Konflic, Prev-Am, Canelys, Oleorgan, Algasil, Ovipron Top, Deffort and BactoSpeine DF) and five conventional (Calypso, Mospilan, Actara, Movento and Karate Zeon). A total of 42 plum shoots/repetition of the untreated trees of the ‘Stanley’ variety were taken from the field in mid/late June, when the aphids’ flight curve was highest. The shoots were chosen to have a similar aphid population of around 100 individuals per shoot, and then three shoots were placed in glass containers with water according to the product being tested (Figure 4). Each sample was labeled according to the applied product, and observations were made 24/48 h after spraying to measure the mortality rate of aphids at each repetition. The treatment was applied until the shoots were evenly moistened (~350–400 mL preparation/variant) using a hand pump sprayer. The laboratory tests were performed under the same conditions (22–23 °C and 45–50% humidity) for the entire period of the study.

### 4.4. Field Testing of the Ecological and Chemical Products 

Similar to laboratory testing, the thirteen phytosanitary products were evaluated in the field for three consecutive vegetative periods with three repetitions/year for each variant. The products were applied in mid/late June on 5–6 annual shoots of a similar vigor and aphid population (approx. 100 individuals per shoot) on six trees of the ‘Stanley’ plum variety/per repetition (Figure 5). The field testing was applied in the morning, when the temperature did not exceed 25 °C and the humidity was about 55–60%, in the same plum crop for the entire period of the study. All spraying was adapted and correlated with the weather conditions and applied with a hand pump sprayer. The amount of preparation (Ecological/Chemical product) applied for each individual variant in the field was identical to that applied in the laboratory trials (350–400 mL).

Before treatment, the trees were labeled according to the product used. The mortality rate of aphids was evaluated 24/48 h after the treatment by visually inspecting the shoots. A magnifying glass was used to facilitate the counting of surviving aphids. The aphid mortality rate was reported as a percentage.

### 4.5. Statistical Analysis

Statistical analysis was performed to determine the (a) differences between the tested ecological products registered 48 h after treatment application; (b) differences between the tested chemical products registered 48 h after treatment application; (c) differences between the two periods of time after the application to establish the impacts of treatments (24/48 h) on ecological variants; (d) efficiency of ecological products compared with the chemically synthesized ones.

The data were analyzed using the XLSTAT by Addinsoft software (version 2019.3.2) [33], which utilizes the MS Office Excel Professional Plus 2019 platform. All data collected from both laboratory and field were arranged in completely randomized blocks, and then the XLSTAT program was used to perform the analysis of variance (ANOVA) [34]. Afterwards, the Duncan’s Multiple Range Test was used to analyze the differences between the different variants [35] at *p* < 0.0001. To test whether there were differences between the ecological products in terms of mortality rate at 24 h and 48 h after treatment over the three experimental years, a two-sample *t*-test [36] was conducted using the XLSTAT application, with a significance level alpha of 0.05.

## 5. Conclusions

The present research conducted over a three-year (2019–2021) vegetative period revealed that two out of the eight ecological products tested (Ovipron Top and Prev-Am) had a high level of efficacy in aphid control in plum crops. This finding is also supported by statistical analysis, which confirmed the efficacy of these two eco-friendly products. Moreover, the highest effect of the two oil-based ecological products (Ovipron Top and Prev-Am) can be observed in just 24 h after applying the treatments, according to statistical analysis. This fact makes these products a very good tool for quickly decreasing the aphid population in plum crops.

The results of this study provide a significant contribution regarding aphid control under ecological system conditions in plum species and fill a gap due to the current lack of information in the specialized literature. Furthermore, this research also provides valuable insights and contributions for related phytosanitary pest control programs in ecological farms, with the scientifically proven efficacy of certain insecticidal products which can be ideal tools for plum farmers.

## Figures and Tables

**Figure 1 plants-12-03316-f001:**
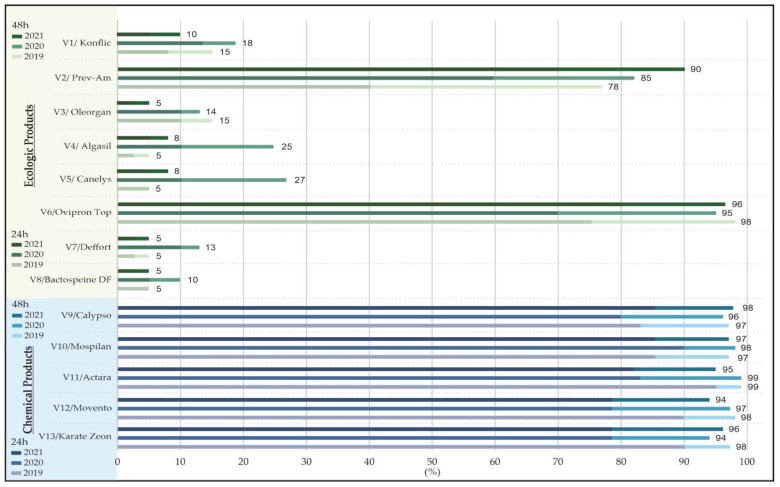
Efficacy of 13 ecological and chemical products in controlling aphids tested under laboratory conditions.

**Figure 2 plants-12-03316-f002:**
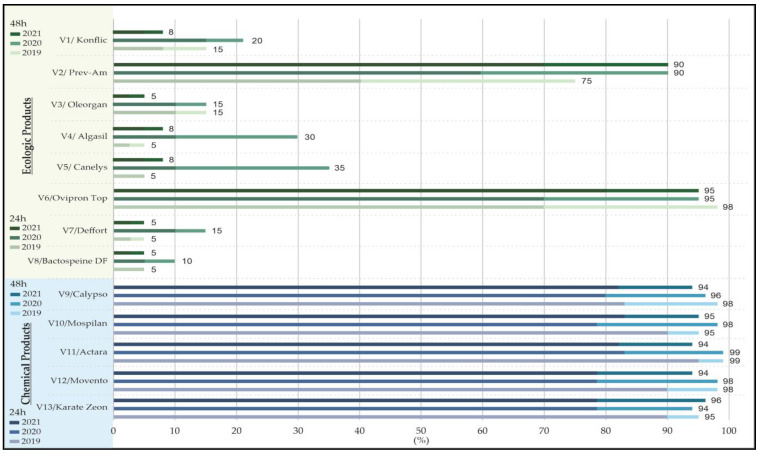
Efficacy of ecological and chemical products in controlling aphids tested in the field.

**Figure 3 plants-12-03316-f003:**
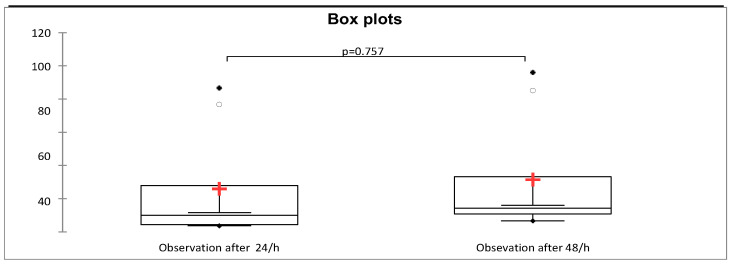
Box plot analysis of two time periods observations (24/48 h).

**Figure 4 plants-12-03316-f004:**
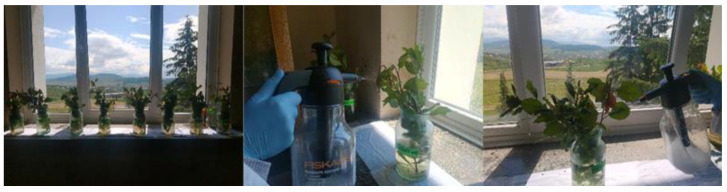
Evaluation of ecological phytosanitary products in the laboratory.

**Figure 5 plants-12-03316-f005:**
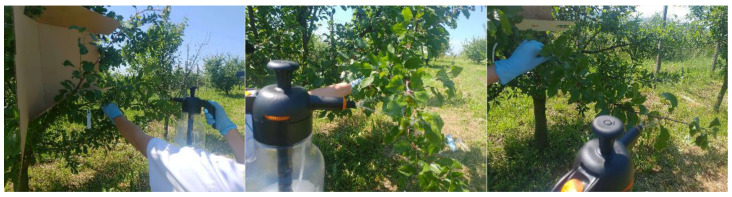
Evaluation of ecological phytosanitary products in the field.

**Table 1 plants-12-03316-t001:** Differences between the tested ecological products registered 48 h after treatment application during the three-year experimental period (2019–2021).

Products	Mortality Laboratory	Mortality Field
Ovipron Top	96.333 ± 1.25 a	96.000 ± 1.41 a
Prev-Am	86.667 ± 4.92 a	85.000 ± 7.07 a
Canelys	16.333 ± 9.74 b	16.000 ± 13.49 b
Konflic	16.333 ± 3.30 b	14.333 ± 4.92 b
Algasil	14.667 ± 8,81 b	14.333 ± 11.15 b
Oleorgan	12.333 ± 5.25 b	11.667 ± 4.71 b
Deffort	8.667 ± 3.77 b	8.333 ± 4.71 b
BactoSpeine DF	7.000 ± 2.36 b	6.667 ± 2.36 b
Pr > F(Model)	<0.0001	<0.0001
Significant	Yes	Yes

Note: Values presented in the table are the averages for every treatment variant, both in the field and in the laboratory. Means followed by different letters indicate statistical differences at *p* < 0.0001 according to Duncan’s Multiple Range Test.

**Table 2 plants-12-03316-t002:** Chemical products, with an analysis of the differences between the categories with a confidence interval of 99% (mortality).

Contrast	Difference	Standardized Difference	Critical Value	Pr > Diff	Alpha (Modified)	Significant
Actara vs. Calypso	1.745	0.964	3.526	0.865	0.039	No
Actara vs. KarateZeon	0.856	0.473	3.469	0.963	0.030	No
Actara vs. Mospilan	0.698	0.386	3.385	0.922	0.020	No
Actara vs. Movento	0.689	0.381	3.250	0.712	0.010	No
Movento vs. Calypso	1.056	0.583	3.469	0.935	0.030	No
Movento vs. KarateZeon	0.167	0.092	3.385	0.995	0.020	No
Movento vs. Mospilan	0.009	0.005	3.250	0.996	0.010	No
Mospilan vs. Calypso	1.047	0.578	3.385	0.835	0.020	No
Mospilan vs. KarateZeon	0.158	0.087	3.250	0.932	0.010	No
KarateZeon vs. Calypso	0.889	0.491	3.250	0.635	0.010	No

**Table 3 plants-12-03316-t003:** Differences between the two time periods (24/48 h) during 2019–2021 after the application of the t-test for two strings of independent data.

Summar Statistics	Data
Difference	−5.625
t (Observed value)	−0.315
|t| (Critical value)	2.145
DF	14
*p*-value (Two-tailed)	0.757
Alpha	0.05

**Table 4 plants-12-03316-t004:** Statistical outcomes of aphid mortality rates 48 h post-treatment for all products tested in this study in the field.

Products	Mortality
Actara	97.333 ± 2.35 a
Mospilan	96.667 ± 1.24 a
Movento	96.667 ± 1.88 a
KarateZeon	96.333 ± 1.24 a
Ovipron Top	96.000 ± 1.41 a
Calypso	95.667 ± 1.24 a
Prev-Am	85.000 ± 7.07 a
Canelys	16.000 ± 13.45 b
Konflic	14.333 ± 4.92 b
Algasil	14.333 ± 11.15 b
Oleorgan	11.667 ± 4.71 b
Deffort	8.333 ± 4.71 b
BactoSpeine DF	6.667 ± 2.36 b
Pr > F(Model)	<0.0001
Significant	Yes

Note: The values presented in the table are averages for every treatment variant. Averages followed by different letters indicate differences at *p* < 0.0001 according to Duncan’s Multiple Range Test.

**Table 5 plants-12-03316-t005:** Details regarding the treatments, concentrations and active substances of the ecological products tested in the study.

Treatment/Product	Concentration (%)	Active Substance
V1/Konflic	0.3%	(50%) Potassium Salt and (50)% Quassia extract
V2/Prev-Am	0.8%	Mineral orange oil 60 g/L
V3/Oleorgan	0.3%	Neem extract 400 g/L
V4/Algasil	0.5%	Algae extract plus K_2_O 90 g/L and SiO_2_ 200 g/L
V5/Canelys	0.3%	Cinnamon extract (70%)
V6/Ovipron Top	2.5%	Mineral paraffinic oil 96.5 g/kh
V7/Deffort	0.3%	Fabaceae family extract 8 g/L
V8/BactoSpeine DF	0.1%	54% *Bacillus thuringiensis*, subsp Kurstaki ABTS 351

**Table 6 plants-12-03316-t006:** Variants of conventional product treatments, their concentrations and active substances tested in the study.

Treatment/Product	Concentration (%)	Active Substance
V1/Calypso	0.02%	Thiacloprid 480 g/L
V2/Mospilan	0.02%	Acetamiprid 200 g/kg
V3/Actara	0.01%	Acetamiprid 200 g/kg
V4/Movento	0.19%	Spirotetramat 100 g/L
V5/Karate Zeon	0.015%	Lambda-cyhalothrin 50 g/L

## Data Availability

The data presented in this study are available in the article.

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
