# Peer review of "Alternative Ecological Products for Aphid Control on Plum"

_plants, 2023, doi:10.3390/plants12183316_

Round 1
Reviewer 1 Report
ID plants-2537181-peer-review-v1
Alternative Ecological Products for Aphids Control on Plum
1. Please provide in the Introduction a paragraph related to volatile organic compounds. There is no paragraph dealing with odors, semiochemicals, plant-insect communication. This is a real alternative to chemical products. Moreover, connected with Ecology. See suggested references below. 2. It is hard to read Tables 1-4. I recommend to prepare Figures with combined Tables 1-2 and 3-4. 3. Is it of importance to present Tables 7 and 8 or 9? Describe Tables in the text. In present form they provide very poor information.
4. Please propose 2-4 Conclussions in this manuscript.
Some other paper for consideration:
Effect of phenolic acid content on acceptance of hazel cultivars by filbert aphid Plant Protection Science 55(2): 116-122 (2019) DOI: 10.17221/150/2017-PPS
Sitophilus granarius responses to blends of five groups of cereal kernels and one group of plant volatiles Journal of Stored Products Research 63: 63-66 (2015) DOI: 10.1016/j.jspr.2015.05.007
Looks acceptable.
Author Response
Dear reviewer,
we thank you very much for the valuable comments and suggestions which allowed us to improve our manuscript. All the changes according to the suggestions received, has been highlighted in the text with yellow in the main document.
- Please provide in the Introduction a paragraph related to volatile organic compounds. There is no paragraph dealing with odors, semiochemicals, plant-insect communication. This is a real alternative to chemical products. Moreover, connected with Ecology. See suggested references below.
As suggested by the reviewer, we added the following sentence to the main document:
“The allelopatic effect of phenolic acids can be also a good tool for protect plants against aphids infestation, the high concentration of phenolic acid (gallic acid and caffeic acid) inflict the reduction of aphid infestations [17]. Another alternative method to decrease the aphid infestation can be the applying of organic blends of volatiles which can attract the pest to be captured [18].” line 59-63
Reference:
- Gantner, M.; Najda, A.; Piesik, D. Effect of phenolic acid content on acceptance of hazel cultivars by filbert aphid. Plant Protection Science, 55(2), 2019, 116-122. [CrossRef], line 361-362
- Piesik, D.; Wenda-Piesik, A. Sitophilus granarius responses to blends of five groups of cereal kernels and one group of plant volatiles. Journal of Stored Products Research, 2015, 62, 36-39. [CrossRef], line 363-364
- It is hard to read Tables 1-4. I recommend to prepare Figures with combined Tables 1-2 and 3-4. 3.
As suggested by the reviewer, we combined Tables 1-2 and 3-4 in to figure 1 and figure 2.
- Is it of importance to present Tables 7 and 8 or 9? Describe Tables in the text. In present form they provide very poor information.
According to the reviewers note we deleted table 3, but the other tables (4 and 5) remain, because it contain the results from the t-test for two strings of data regarding the two time periods (24/28 hours) during 2019-2021 with the graphically presented boxplot (table 4), respectively table 5 which is important and contain information regarding the statistical outcomes of aphid mortality rate 48 h post treatment for all products tested in the study, organized according statistical ranking (“a” and “ b” letters) aimed by Duncan`s Multiple Range Test.
- Please propose 2-4 Conclussions in this manuscript.
As suggested by the reviewer, we added new conclussions based of results obtained in current stage:
“Moreover the highest effect of the two oil-based ecological products can be observed in just 24 hours after applying the treatments, according to statistical analysis. This fact makes these products a good tool to decrease quickly the population of aphids in plum crop.” line 308-310
NOTE: The manuscript with the all modifications made is attached below
Author Response
Dear reviewer,
we thank you very much for the valuable comments and suggestions which allowed us to improve our manuscript. All the changes according to the suggestions received, has been highlighted in the text with turquoise in the main document.
- Errors in References. Please check the entire bibliography.
As suggested by the reviewer, the references section was carefully revised and corrected all errors following the rules of the journal.
- The material and methods chapter is written very generally. I am asking you to improve.
As suggested by the reviewer, the materials and methods chapter was improved by adding more detailed information.
- Section 4.3 incorrect title. Chemical preparations were also tested.
As suggested by the reviewer, the title has been changed in “4.3. Laboratory Testing of the Ecologic and Chemical Products” at line 259.
- No number of repetitions
As suggested by the reviewer, the number of repetitions was added in main document in all the places where it was necessary. The study was carried out over three consecutive vegetative periods (2019-2021), in three repetitions per year for all products tested.
- Under what conditions were the experiment conducted, were they constant
As suggested by the reviewer, we added the following sentence to the main document:
“The laboratory tests were performed in the same conditions (22-23°C and 45-50% humidity) on the entire period of the study.” line 271-273
- How much of the preparation was applied to the plant, what equipment was used?
As suggested by the reviewer, we added the following sentence to the main document:
“The laboratory tests performed under the same conditions (22-23°C and 45-50% humidity) for the entire period of the study. The treatment was applied until the shoots were evenly moistened (~350-400 ml preparation/variant), using a hand pump sprayer.” line 270-273.
- Section 4.4. incorrect title. Chemical preparations were also tested.
As suggested by the reviewer, the title has been changed in “4.4. Field Testing of the Ecologic and Chemical Products ” at line 276.
- Please describe in detail the field testing methodology is too general
As suggested by the reviewer, we added the following sentence to the main document:
“The field testing was applied in the morning, when the temperature did not exceed 25°C and the humidity was about 55-60%, in the same plum crop for the entire period of the study. All spraying was adapted and correlated with the weather conditions and applied with a hand pump sprayer. The amount of preparations (Ecologic/Chemical product) applied for each individual variant in the field was identical to that applied in the laboratory trials (350-400 ml).” line 281-286.
NOTE: The manuscript with all changes is attached below.
Reviewer 3 Report
This paper is focusing to evaluate eight ecological products in comparing with five chemical products toward aphids on plum tree, but only two ecological products (Ovipron Top and PrevAm) has a positive results.
Actually, the paper looks very well written and clearly described, for that I would like to congratulate the authors for their efforts.
Author Response
Thank you so much.
Round 2
Reviewer 1 Report
Thank you for providing corrected version of the manuscript.
Reviewer 2 Report
There are no comments.